# Acylated heptapeptide binds albumin with high affinity and application as tag furnishes long-acting peptides

Alessandro Zorzi[1], Simon J. Middendorp[1], Jonas Wilbs[1], Kaycie Deyle[1] & Christian Heinis[1]

The rapid renal clearance of peptides *in vivo* limits this attractive platform for the treatment of a broad range of diseases that require prolonged drug half-lives. An intriguing approach for extending peptide circulation times works through a 'piggy-back' strategy in which peptides bind via a ligand to the long-lived serum protein albumin. In accordance with this strategy, we developed an easily synthesized albumin-binding ligand based on a peptide-fatty acid chimera that has a high affinity for human albumin ($K_d = 39$ nM). This ligand prolongs the elimination half-life of cyclic peptides in rats 25-fold to over seven hours. Conjugation to a peptide factor XII inhibitor developed for anti-thrombotic therapy extends the half-life from 13 minutes to over five hours, inhibiting coagulation for eight hours in rabbits. This high-affinity albumin ligand could potentially extend the half-life of peptides in human to several days, substantially broadening the application range of peptides as therapeutics.

[1] Institute of Chemical Sciences and Engineering, School of Basic Sciences, Ecole Polytechnique Fédérale de Lausanne (EPFL), Lausanne CH-1015, Switzerland. Correspondence and requests for materials should be addressed to C.H. (email: christian.heinis@epfl.ch).

Peptides combine a range of favourable qualities such as suitable affinity and selectivity, low inherent toxicity, and access to chemical synthesis which make them an attractive modality for therapeutic development[1]. Many peptides with desired binding specificities and biological activities have been identified in organisms such as bacteria, fungi or plants. Peptides with tailored binding specificities can also be developed by *in vitro* evolution techniques such as phage display or mRNA display[2,3]. The modularity of peptides, the large number of commercially available amino acid building blocks, and their compatibility with automated synthesis allow for fast development from lead structures to high-affinity ligands.

A major obstacle remaining for developing peptides as therapeutics is their rapid clearance from the circulation due to proteolytic degradation and/or fast renal filtration through large pores of the glomeruli in the kidney. This limits the development of peptide drugs that need to block targets over extended time periods such as enzyme inhibitors or receptor antagonists. While strategies exist for improving the stability of peptide leads, the fast renal clearance remains a great challenge. Not surprisingly, most of the approved peptide drugs are agonists that act in minutes and thus do not need to linger in the body. To give some indication of the fast renal clearance of proteolytically stable peptide drugs, insulin, once in the blood stream has a half-life of 4–6 min[4]. The hormone drug oxytocin has a half-life of 10–15 min when applied intravenously[5].

A 'piggy-back' strategy provides an attractive approach for prolonging the residence times of peptides in circulation. In this method, peptides are conjugated to ligands that bind serum proteins, such as albumin, with long plasma half-lives[6,7]. Albumin, for example, has a half-life of 19 days in humans, largely due to its size above the renal filtration threshold and its unique ability to interact with the neonatal FcRn receptor. It is the most abundant protein in plasma, present at 40 mg ml$^{-1}$ (600 µM)[8]. Three peptide drugs containing albumin-binding fatty acids in their structures, insulin detemir, insulin degludec and liraglutide, are already used in the clinic[9,10]. Insulin detemir is an insulin mutant to which the albumin-binding ligand myristic acid was appended. The drug binds albumin with a $K_d$ of 4.2 µM (ref. 11). Adding the fatty acid slows absorption of insulin from the subcutaneous injection site, and its affinity for albumin extends the circulating half-life. Conjugation of the same insulin mutant via a glutamic acid spacer to a fatty di-acid yielded the drug insulin degludec which has a 2.4-fold higher affinity for albumin and a further prolonged half-life[12]. Liraglutide is a GLP-1 peptide with a prolonged half-life due to the incorporation of a palmitic acid moiety[13]. Despite the attraction of fatty acid conjugation, the relatively weak affinity for albumin and the low solubility of lipidated polypeptides in physiological solutions can be significant limitations[14]. Small molecule albumin ligands resembling the structures of fatty acids were recently identified through compound library screening[15,16]. Alternatively to small molecules, peptide and protein-based albumin ligands were developed[7]. An albumin-binding 18-amino acid peptide developed by phage display, SA21 efficiently prolonged the half-life and the stability of a range of peptides in animal studies[17,18]. Limitations of the SA21 albumin-binding peptide include low solubility caused by the presence of many hydrophobic amino acids, its complex synthesis resulting from the long sequence, and the presence of a disulfide bridge[17].

In this work, we set out to develop an albumin-binding ligand more in line with our needs. This ligand would take into account previously mentioned limitations in areas such as albumin-binding properties, solubility, and ease of synthesis. Specifically, we aimed at engineering a ligand that has a high affinity for albumin with a $K_d$ in the medium to low nanomolar range, has

high solubility in physiological solutions, ideally at least 1 mg ml$^{-1}$ in PBS, and can efficiently be synthesized in conjunction with therapeutic peptides. Towards this end, we conceived a ligand design in which a fatty acid is conjugated to a short peptide via an amino acid side chain. The fatty acid alone should bind to albumin with an affinity in the low micromolar range and the peptide moiety should enhance this affinity by forming additional contacts to albumin. Polar or charged amino acids, which would be favoured in the screening process, should also help to increase the solubility. The peptide-fatty acid format provides for an efficient synthesis, as it can be added robotically during the solid phase synthesis of therapeutic peptides. By following an approach based on iterative rounds of modification of the peptide moiety and affinity testing by fluorescence polarization, we indeed identified a high-affinity albumin ligand that substantially prolongs the *in vivo* half-life of peptides.

## Results

**Synthesis of peptides carrying a peptide-fatty acid tag.** We developed a strategy for the automated synthesis of peptides carrying an albumin ligand of the format shown in Fig. 1a on a standard peptide synthesizer (Fig. 1b). The amine side chain of

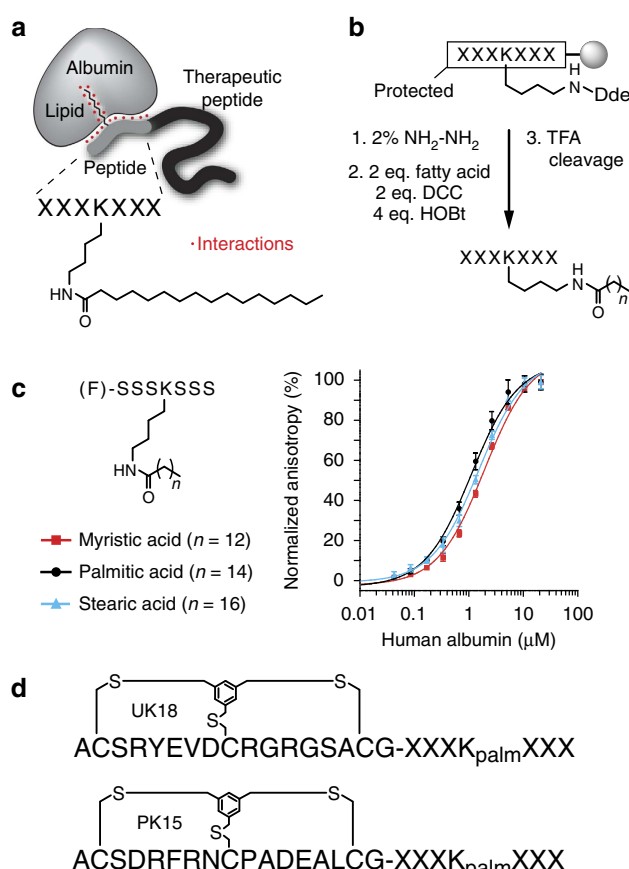

**Figure 1 | Albumin-binding ligand based on a chimeric peptide-fatty acid structure.** (**a**) Peptide-fatty acid tag is appended to either end of a therapeutic peptide. Amino acids X strengthen the binding of the fatty acid that is linked via its carboxylic group to the side chain of a lysine. (**b**) Synthesis of peptide-fatty acid tag on solid phase. (**c**) Binding of chimeric peptide-fatty acid structures to albumin measured by fluorescence polarization. The peptide is modified at the N-terminus with fluorescein (F). Average values and s.d.'s of three measurements are shown. (**d**) Bioactive bicyclic peptides tagged with palmitoylated peptide. palm = palmitoyl, X = Gly or Ser.

lysine (K) in the peptide sequence XXXKXXX (where X is any amino acid) was protected with Dde (1-(4,4-dimethyl-2,6-dioxycyclohex-1-ylidene)ethyl) for selective deprotection with 2% hydrazine. The fatty acid was conjugated via the carboxylic acid using dicyclohexylcarbodiimide (DCC) as an activating reagent. Typical yields for a synthesis on a small scale (for example, 0.03 mmol) were 60–70% after purification.

To confirm that the peptide-fatty acid format binds albumin with a low micromolar $K_d$, we synthesized heptapeptides of the sequence (F)-SSSKSSS-NH$_2$ (where F is fluorescein) and conjugated one of three fatty acids to the lysine side chain (Supplementary Figs 1 and 2). A fluorescein on the N-terminus enabled the measurement of the binding affinity for albumin by fluorescence polarization. Heptapeptides carrying a myristic, palmitic or stearic fatty acid bound to human albumin with $K_d$s of $1.9 \pm 0.2$, $1.08 \pm 0.12$ and $1.48 \pm 0.15\,\mu M$, demonstrating comparable affinities to the therapeutic peptides insulin and GLP-1 acylated with fatty acids (Fig. 1c)[11,12]. Similar affinities were measured when the binding assay was performed in human serum (Supplementary Fig. 3). Although all of the fatty acids demonstrated similar $K_d$s for albumin, palmitic acid had the highest affinity and was the only one carried forward to future tests. This fatty acid is abbreviated $K_{palm}$ when conjugated to lysine in the peptide sequence.

We then tested to what extent the peptide-palmitoyl tag reduces the solubility of peptides. As described above, poor solubility has been a major limitation of peptides tagged with fatty acids or the albumin-binding peptide SA21 (refs 14,18). We synthesized the two well-characterized phage display-derived bicyclic peptides UK18 (ref. 19) and PK15 (ref. 20) as fusions with the peptide-palmitoyl tag GGGK$_{palm}$GGG or SSSK$_{palm}$SSS (Fig. 1d and Supplementary Fig. 4). The solubility in PBS dropped from $> 10\,mg\,ml^{-1}$ for both UK18 and PK15 to 0.22 and 0.68 $mg\,ml^{-1}$, respectively, with the GGGK$_{palm}$GGG tag and to 1.83 and 2.96 $mg\,ml^{-1}$, respectively, with the SSSK$_{palm}$SSS tag. The large difference between the glycine- and serine-based tags indicated that the solubility is greatly influenced by the amino acid composition of the peptide and that it might be improved with suitable sequences.

**Improving the affinity of the heptapeptide-palmitoyl tag**. To identify an amino acid sequence for the heptapeptide-palmitoyl tag that improves both the binding affinity for albumin and the solubility in aqueous solution, we devised the following iterative synthesis and screening approach. Variants of the lead structure (F)-SSSK$_{palm}$SSS-NH$_2$ were synthesized wherein one of the six serine residues at a time was substituted to either W, Y, L, D, E or G (Fig. 2a). These amino acids were chosen to the exclusion of the others because the presence of both hydrophobic and negatively charged groups are basic structural requirements for ligand binding to albumin[21]. The binding affinity for each of these peptides for human albumin was measured in triplicate by fluorescence polarization. The s.d.'s generally fell below 15% of the $K_d$ value which allowed detection of relatively small affinity improvements. The variant with the best affinity for albumin was selected for the next cycle of amino acid substitution and affinity testing. If several amino substitutions showed a similar affinity improvement, a change to D or E was preferred as a charged amino acid could enhance the solubility. All results being equal, E was chosen due to the propensity of D to isomerize to iso-Asp or to racemize to D-Asp. If different positions provided similar affinity improvements, the one closest to the lysine was fixed in order to evolve the peptide from the central palmitoyl anchor to the peptide ends. The iterative process of tag synthesis and affinity screening was repeated until optimal residues were found for all serine positions.

In the first cycle of synthesis and screening, six out of the 36 tag variants, all with D or E amino acid replacements, showed an approximately twofold improvement in binding affinity (greater than twofold improved $K_d$s are highlighted in dark blue in Fig. 2a). We fixed E in position five (fixed substitutions carried through to further cycles are highlighted in red in Fig. 2a). In the second cycle, six of the 30 variants of (F)-SSSK$_{palm}$ESS-NH$_2$ showed an approximately twofold improvement in binding affinity again in the substitutions to D or E. We fixed E in position three and varied the remaining four serine positions. In the third cycle, six out of the 24 variants of (F)-SSEK$_{palm}$ESS-NH$_2$ had an approximately twofold improved affinity, wherein three were substituted to W or Y, and three to D or E. We followed two paths for further optimizations, one by fixing E in position two (cycle 4a) and one by fixing Y in position two (cycle 4b). Following path 4a, no large improvements could be achieved. In contrast, path 4b enhanced the binding affinity by nearly another factor two when any of the three serine positions were occupied by E. We subsequently followed three paths (cycles 5a–c) and substituted the remaining two serine positions to Y or E. Two of the paths converged to the same best binder, delivering the sequence (F)-EYEK$_{palm}$ESE-NH$_2$. In the last serine position, Y gave a slightly better affinity than E. This final ligand EYEK$_{palm}$EYE had a $K_d$ of $39 \pm 3\,nM$ and was thus 27-fold improved compared to the starting peptide-palmitoyl tag. The evolved amino acid sequence did not share any homology with binding regions of natural albumin-binding proteins.

The affinities of the best tag variants from each cycle were measured on the same assay plate using the same dilutions of albumin to exclude inaccuracies rising from the separate preparation of albumin solutions (Fig. 2b). The graphic shows that the binding was improved by around a factor two in the first rounds of evolution, and that affinity enhancements decreased in the last cycles as we sharpened the affinity to the final peptide sequence. Transferring the fluorescein in the final peptide EYEK$_{palm}$EYE from the N- to the C-terminus reduced the binding affinity several fold, which suggested that the fluorescein moiety is participating in the binding to albumin when attached to the N-terminus. In the following characterization studies and applications, fluorescein is conjugated to the N-terminus, and the evolved peptide-palmitoyl tag (F)-EYEK$_{palm}$EYE is abbreviated to 'tag'.

In blood, fatty acids bound to albumin could potentially compete with the binding of the palmitoylated heptapeptide. Incubation of the evolved tag and best variants with human serum showed efficient binding in the context of fatty acids and other blood components (Fig. 3a and Supplementary Fig. 5). To apply the tag in pre-clinical studies, it is necessary that it binds to the albumin of other species. Fluorescence polarization assays with rat and rabbit albumin showed that the tag binds to the orthologous albumins with lower affinity (rat: $K_d = 220 \pm 30\,nM$, rabbit: $K_d = 320 \pm 40\,nM$). Analysis of the best tag variants in the same assay showed that the binding to rat albumin was also improved in each evolution cycle but to a smaller extent (Fig. 3a and Supplementary Fig. 5).

**Application of the evolved tag to bioactive peptides**. We applied the tag to the bicyclic peptide PK128, an inhibitor of plasma kallikrein ($K_i = 300\,pM$)[22] and to the bicyclic peptide UK18, an inhibitor of urokinase ($K_i = 53\,nM$)[19]. Both protease inhibitors suffered from fast renal clearance upon previous tests in mice[18]. In addition, PK128 has a poor stability in plasma ($t_{1/2}$ human plasma = $1.8 \pm 0.3\,h$). We synthesized the two peptides by appending the tag (F)-EYEK$_{palm}$EYE to the peptides' N-termini during SPPS, spaced by a short polyethylene glycol (PEG) linker.

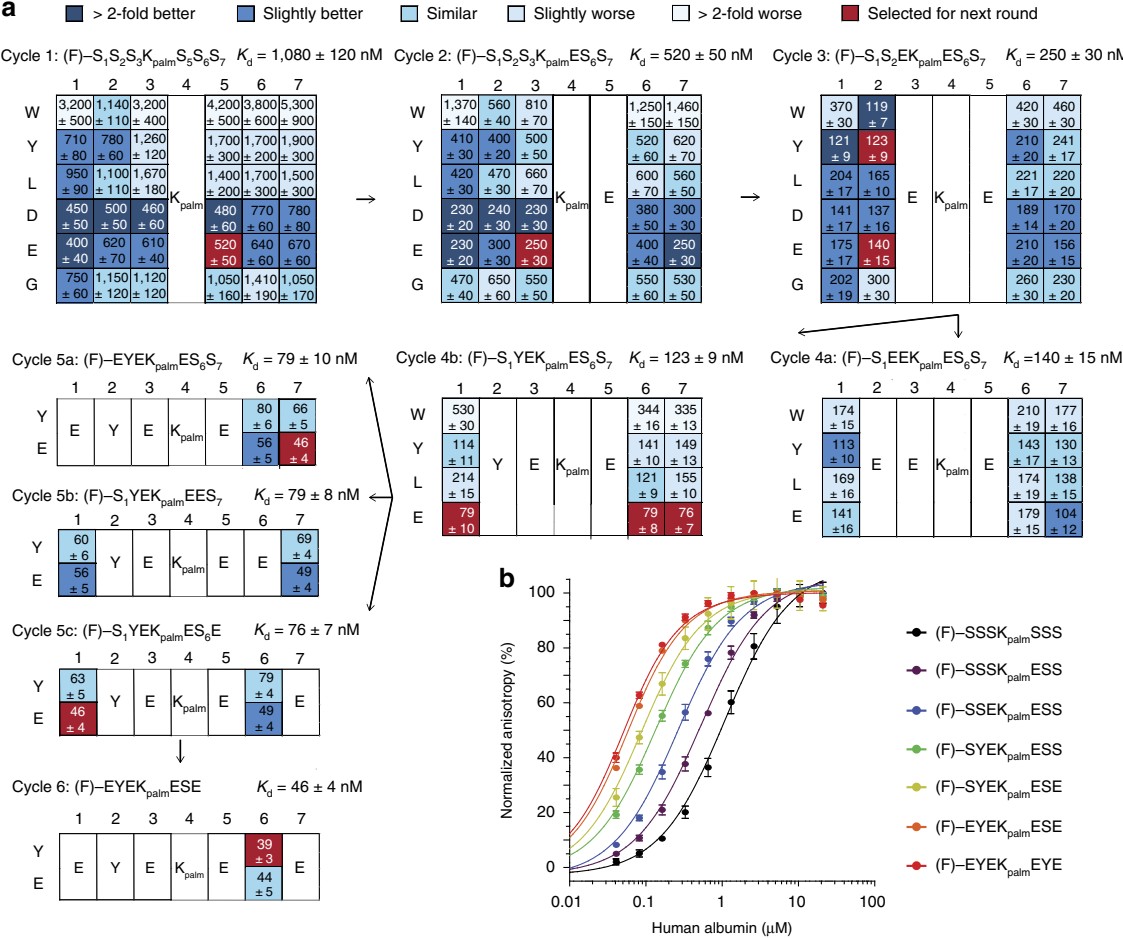

**Figure 2 | Evolution of peptide moiety of the tag.** (**a**) Six cycles of iterative heptapeptide-palmitoyl tag synthesis and affinity measurement are shown. For each cycle, the sequence of the starting peptide (for example, (F)-$S_1S_2S_3$ $K_{palm}S_5S_6S_7$) is shown together with its dissociation constant $K_d$ for human albumin. One serine was substituted at the time to the indicated amino acids (W, Y, L, D, E and G) and the affinity of the different heptapeptide-palmitoyl variants are indicated in the boxes. Dark blue or red shaded boxes indicate variants with the best binding affinities. Variants in boxes highlighted in red were used for the next evolution cycle. (**b**) Binding of the best tag variants evolved in the different cycles measured by fluorescence polarization. Average values of at least three measurements and s.d.'s are indicated.

Fluorescein was included in the tag for binding purposes as well as for the facile measurement of the $K_d$ by fluorescence polarization and the detection of the tagged peptides in blood samples. A single step of purification by high-performance liquid chromatography (HPLC) provided a 40% yield of tag-PK128 and tag-UK18 with >95% purity, reconfirming the high efficiency of the synthesis strategy (Fig. 3b and Supplementary Fig. 6). The two conjugates showed a solubility in PBS of $1.23 \pm 0.35$ and $1.18 \pm 0.18$ mg ml$^{-1}$, respectively.

The two tagged bicyclic peptides bound to human albumin with four to fivefold weaker dissociation constants than the tag alone ($K_d$ tag-PK128 = $194 \pm 16$ nM, $K_d$ tag-UK18 = $168 \pm 13$; compared to $K_d$ tag = $39 \pm 3$ nM; Table 1 and Supplementary Fig. 7). As expected, a control with non-tagged UK18 showed no binding to albumin. As with the tag alone, the tagged bicyclic peptides displayed a similar affinity to human albumin when measured in serum compared to the purified protein (Table 1). The binding of the tagged bicyclic peptides to rat albumin decreased three to fourfold compared to the tag alone and was overall around fivefold weaker than the binding to human albumin (Table 1). Both of the two tagged bicyclic peptides inhibited the protease targets with $K_i$s in the nanomolar range, tag-PK128 plasma kallikrein with a $K_i$ of $9.7 \pm 0.8$ nM and tag-UK18 urokinase with a $K_i$ of $56 \pm 15$ nM (Table 1 and

Supplementary Fig. 8). The inhibitory activity of tagged PK128 was thus 30-fold weaker and that of tagged UK18 was unchanged. In the presence of albumin (25 μM), the inhibitory activity of tag-PK128 was further reduced ($K_i = 79 \pm 18$ nM) while tag-UK18 remained as active as the inhibitor without tag ($K_i = 80 \pm 6$ nM). It is likely that the tag sterically hinders binding of PK128 to plasma kallikrein to some extent and that this effect is enhanced when it is bound to albumin.

The tagged bicyclic peptides were resistant to plasma proteases as assessed by the measurement of the residual inhibition activity of tag-PK128 and tag-UK18 after incubation in human plasma at 37 °C for two days (Fig. 3c). The stabilization effect resulting from albumin association was most pronounced for PK128 which retained its full inhibition activity after 2 days in plasma ($t_{1/2} > 48$ h), while non-tagged PK128 is inactivated within hours ($t_{1/2} = 1.8 \pm 0.3$ h).

**Stability and pharmacokinetics of tagged peptides in rats.** We tested the half-life of tag-PK128 and tag-UK18 in rats by i.v. administration of 0.5 mg kg$^{-1}$ followed by HPLC analysis of blood samples ($n = 3$ per peptide; Fig. 4a, Supplementary Figs 9 and 10). For comparison, we measured the half-life of UK18 (0.3 mg kg$^{-1}$) and the tag alone (0.25 mg kg$^{-1}$) ($n = 3$ per

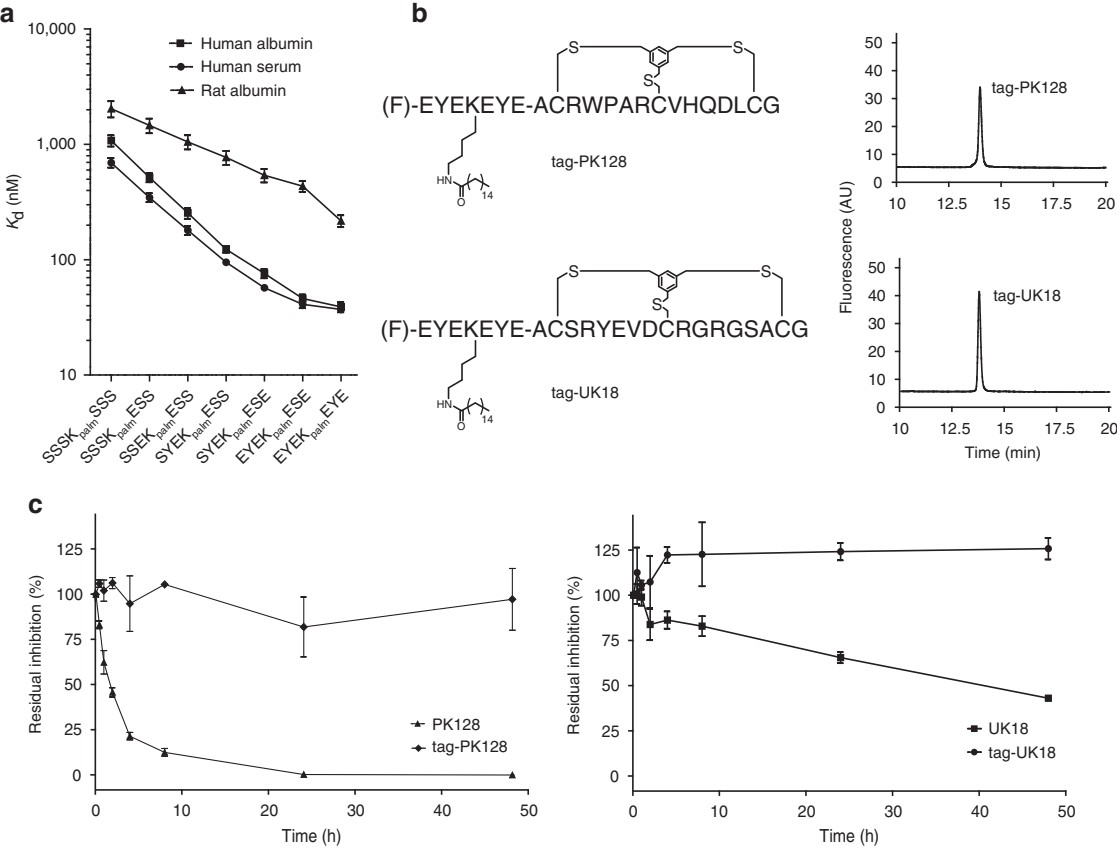

**Figure 3 | Application of evolved tag to bioactive peptides.** (**a**) Binding of peptides from different evolution cycles to human and rat albumin. Average values and s.d.'s of three measurements are shown. (**b**) Structures of tag-PK128 and tag-UK18 and their purity after automated synthesis and a single HPLC purification. (**c**) Stability of tagged and non-tagged bicyclic peptide PK128 and UK18 in human plasma at 37 °C. Residual inhibition activities were quantified in protease activity assays using fluorogenic substrates of plasma kallikrein and urokinase. Average values and s.d.'s of three measurements are shown.

**Table 1 | Characterization of tag-PK128 and tag-UK18.**

| | Albumin binding ($K_d$) (nM) | | | Protease inhibition ($K_i$) (nM) | |
|---|---|---|---|---|---|
| | **Human albumin** | **Human serum** | **Rat albumin** | **No albumin** | **With albumin** |
| tag | $39 \pm 3$ | $37 \pm 2$ | $220 \pm 30$ | n.d. | n.d. |
| PK128 | n.d. | n.d. | n.d. | $0.35 \pm 0.04$ | $0.18 \pm 0.11$ |
| tag-PK128 | $194 \pm 16$ | $181 \pm 5$ | $720 \pm 90$ | $9.7 \pm 0.8$ | $79 \pm 18$ |
| UK18 | $>100,000$ | $>100,000$ | $>100,000$ | $55 \pm 9$ | $34 \pm 7$ |
| tag-UK18 | $168 \pm 13$ | $149 \pm 6$ | $780 \pm 110$ | $56 \pm 15$ | $80 \pm 6$ |

n.d., not determined.
Binding to albumin was measured by fluorescence polarization. For serum assays, a human albumin concentration of 0.6 mM was used to calculate the $K_d$. Inhibition constants were measured in protease activity assays using fluorogenic substrates. $K_i$ values indicated for PK128 and tag-PK128 refer to plasma kallikrein inhibition and $K_i$s for UK18 and tag-UK18 to urokinase inhibition. Inhibition in the presence of albumin was measured using 25 μM human albumin. All measurements were done in triplicate and s.d.'s are indicated.

peptide; Fig. 4a, Supplementary Figs 11 and 12). The tagged bicyclic peptides had elimination half-lives of $2.9 \pm 0.7$ (tag-PK128) and $7.4 \pm 0.2$ h (tag-UK18), meaning they cleared much slower than a non-tagged bicyclic peptide with a half-life of only minutes. For a specific comparison, the non-tagged control peptide UK18 had an elimination half-life of $18 \pm 1$ min indicating that the conjugation prolonged the circulation time 25-fold. The tag alone had a longer elimination half-life ($12.3 \pm 0.4$ h) than the tagged bicyclic peptides, most likely due to its slightly higher affinity for albumin. Detailed pharmacokinetic parameters including distribution half-life, clearance rate and volume of distribution are shown in Table 2. The HPLC analysis of blood samples revealed partial degradation of UK18 (Fig. 4b, Supplementary Figs 11 and 13). For the conjugates tag-PK128

and tag-UK18, no degradation products could be detected, which again indicates the protective effect of the conjugation (Supplementary Figs 9 and 10). Despite its linear format, the tag alone was not degraded, most likely because all amino acids of the peptide interact with albumin or are near its surface and thus are not accessible to proteases (Supplementary Fig. 12).

**Application of the evolved tag to a FXIIa inhibitor.** Several recent studies indicate that coagulation factor XII (FXII) plays a key role in various thrombotic disorders and have featured the serine protease as a therapeutic target[23]. Our laboratory has developed a selective bicyclic peptide inhibitor of activated FXII (FXIIa; FXII618)[24] that was recently optimized to a picomolar

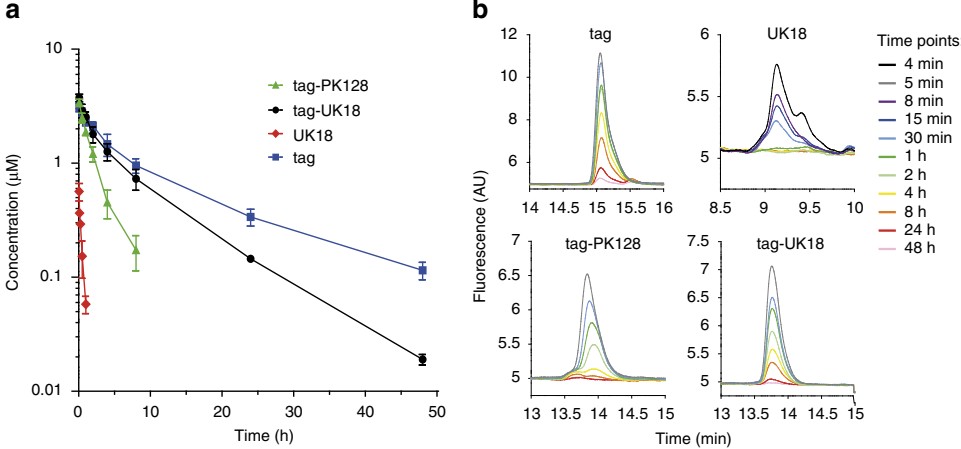

**Figure 4 | Pharmacokinetic study in rats.** (**a**) Concentration of peptides in rat plasma at indicated time points after i.v. administration of $0.5\,mg\,kg^{-1}$ tagged bicyclic peptides, $0.3\,mg\,kg^{-1}$ UK18 or $0.25\,mg\,kg^{-1}$ tag ($n = 3$). s.d.'s are indicated. (**b**) Analysis of peptides in rat blood samples by RP-HPLC using a fluorescence detector.

**Table 2 | Parameters of pharmacokinetic study in rats.**

|  | tag-PK128 | tag-UK18 | UK18 | tag |
|---|---|---|---|---|
| $t_{1/2}\,\alpha$ (h) | $0.9 \pm 0.2$ | $1.0 \pm 0.1$ | n.d. | $1.5 \pm 0.4$ |
| $t_{1/2}\,\beta$ (h) | $2.9 \pm 0.7$ | $7.4 \pm 0.2$ | $0.30 \pm 0.02$ | $12.3 \pm 0.4$ |
| AUC ($mg\,min\,ml^{-1}$) | $1.7 \pm 0.3$ | $4.6 \pm 0.6$ | $0.034 \pm 0.004$ | $3.2 \pm 0.5$ |
| CL ($ml\,h^{-1}\,kg^{-1}$) | $18 \pm 4$ | $7 \pm 1$ | $500 \pm 70$ | $4 \pm 1$ |
| $V_{D}$ ($ml\,kg^{-1}$) | $76 \pm 7$ | $77 \pm 12$ | $200 \pm 40$ | $74 \pm 13$ |

AUC, area under curve; CL, clearance; n.d., not determined; $V_{D}$, volume of distribution.
$n = 3$ per peptide. Average values and s.d.'s are indicated.

inhibitor ($K_i = 0.84 \pm 0.03\,nM$)[25,26]. For application in antithrombotic therapy, the inhibitor needs to block FXIIa over an extended period of time. To enable prolonged exposure, we applied the tag to the FXIIa inhibitor. Conjugation of the tag to either end of the bicyclic peptide reduced its affinity over 20-fold wherein PEG linkers of various lengths attached between the tag and inhibitor limited the loss in affinity somewhat (Supplementary Table 1, Supplementary Figs 14–16). The most active conjugate was obtained when the tag was linked via a long PEG chain ($(NH(CH_2CH_2O)_{24}CO)_3$) to the N-terminus of the FXIIa inhibitor ($K_i$ FXIIa $= 4 \pm 0.9\,nM$ in presence of albumin, $K_d$ human albumin in serum $= 224 \pm 11\,nM$; Supplementary Figs 17 and 18). This conjugate, termed tag-3xPEG24-FXIIa inhibitor, showed a high target selectivity, inhibiting other plasma proteases only at concentrations that are >1,000-fold higher than the $K_i$ (Supplementary Fig. 19). It showed a remarkable solubility of more than four $mg\,ml^{-1}$ in PBS.

Inhibition of FXIIa in blood can be assessed by measuring the so-called activated partial thromboplastin time, the time until citrated plasma clots upon activation of the intrinsic coagulation pathway. We measured this parameter in rabbit plasma *ex vivo*. At a concentration of $10\,\mu M$, the different conjugates inhibited the activation of the intrinsic pathway efficiently and their activity correlated directly with the FXIIa inhibition constants (Fig. 5a). The best conjugate, tag-3xPEG24-FXIIa inhibitor showed a fivefold prolongation of the coagulation time ($EC_{5x}$) at $4.2 \pm 0.5\,\mu M$ (Supplementary Table 2). The conjugate bound rabbit albumin with a $K_d$ of $1.6 \pm 0.3\,\mu M$ and thus sevenfold weaker than human albumin. Upon i.v. administration of the tag-3xPEG24-FXIIa to rabbits ($n = 3$; $5\,mg\,kg^{-1}$) we found an elimination half-life of $5.2 \pm 0.4\,h$ which was 24-fold longer than

that of non-tagged FXIIa inhibitor ($n = 3$; $3.7\,mg\,kg^{-1}$; $t_{1/2} = 13 \pm 2\,min$) (Fig. 5b, Supplementary Table 3 and Supplementary Fig. 20). With the greatly prolonged circulation time, the intrinsic coagulation pathway could efficiently be inhibited for 8 h (Fig. 5c).

## Discussion

Lipidation is an established strategy for extending the circulation time of peptide drugs as showcased by the approved drugs insulin detemir, insulin degludec and liraglutide. In all of these peptide drugs, fatty acids such as myristic or palmitic acid are appended to an amino acid within the peptide sequence itself. It can be expected that the plasma half-life of these or other peptide drugs could be substantially prolonged by applying the concept of embedding the fatty acid into a peptide sequence that enhances the binding to albumin. Studies on fatty acid-modified insulin analogues and peptide-based albumin binders showed that a stronger affinity for albumin correlated with a prolonged half-life[27,28].

Herein, we demonstrate that a peptide sequence as short as seven amino acids (including the lipidated lysine) can improve the binding of a fatty acid conjugated to a peptide by 27-fold. The peptide was evolved with a simple and efficient strategy based on six rounds of iterative amino acid substitution and affinity measurement, in which more than 150 heptapeptide-palmitoyl tags were tested. As expected, amino acids with negatively charged or hydrophobic side chains were essential for the increased binding affinity, although the specific sequence of these amino acids was also an important factor. Due to the four negative charges in the peptide, the tagged conjugates have a good solubility in aqueous solution, despite the long hydrophobic tail of palmitic acid, preventing solubility issues of the lipidated peptides.

We showed that the tag can be appended conveniently to cargo peptides by automated synthesis on a standard peptide synthesizer. We applied the evolved tag to three bioactive peptides and presented that the conjugates bind with the desired affinity to both the peptides' targets and albumin, and that the cargo peptides have an improved proteolytic stability over the non-tagged peptides. Furthermore, we demonstrated that the tag prolongs the circulation time of the peptides in rats and rabbits around 24-fold, reaching half-lives of 7 and 5 h. Application of the tag to a bicyclic peptide FXIIa inhibitor allowed efficient blockage of the intrinsic coagulation pathway in rabbits as long as

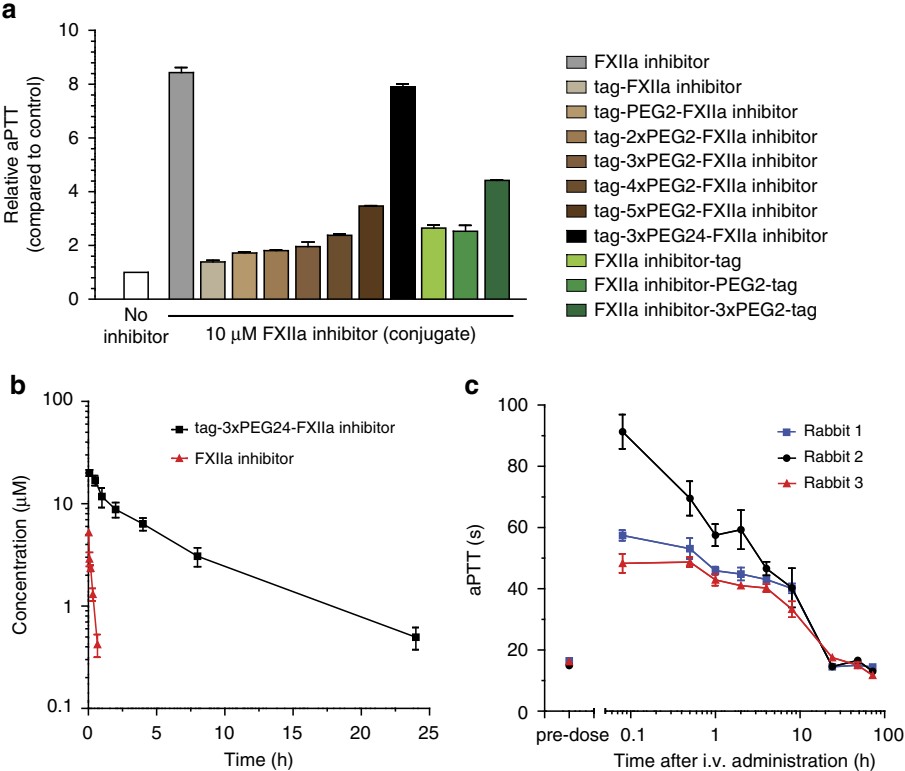

**Figure 5 | Inhibition of the intrinsic coagulation pathway with a tag-FXIIa inhibitor conjugate in rabbits. (a)** aPTT is the time necessary for human plasma to clot upon activation of the intrinsic coagulation pathway. The indicated relative aPTT is the factor by which the coagulation time is prolonged in the presence of 10 μM FXIIa inhibitor or tagged FXIIa inhibitor tested in plasma *ex vivo*. The tag was conjugated via different linkers to either the N- or C-terminus of the bicyclic peptide FXIIa inhibitor. PEG2 and PEG24 stand for linkers containing two or 24 CH$_2$CH$_2$O units. **(b)** Study of pharmacokinetics. Concentration of FXIIa inhibitor in rabbit plasma after i.v. administration of 5 mg kg$^{-1}$ tagged FXIIa inhibitor (tag-3xPEG24-FXIIa inhibitor) or 3.7 mg kg$^{-1}$ FXIIa inhibitor (n = 3). s.d.'s are indicated. **(c)** aPTT in rabbit plasma samples taken before and after i.v. administration of 5 mg kg$^{-1}$ tag-3xPEG24-FXIIa inhibitor. Average values and s.d.'s of three measurements are shown. aPTT, activated partial thromboplastin time.

eight hours. It is important to note that the tag has a weaker affinity for rat and rabbit than for human albumin (five to sevenfold). Taking this difference and allometric scaling considerations into account, it is likely that peptide drugs modified with the tag can achieve half-lives of more than one day in human. If such conjugates are administrated as a depot, an even longer exposure may be reached.

With this tag in hand, it should be possible to expand the application range of peptide therapeutics from the current mostly short-lived agents that act mainly as receptor agonists to long-acting peptide drugs that can also address targets that require actions over extended time periods, such as the inhibition of enzymes or receptors.

## Methods

**Peptide synthesis.** Peptides were synthesized on an Advanced ChemTech 348Ω parallel peptide synthesizer (AAPPTec) by standard Fmoc solid-phase chemistry on Rink Amide resin (0.26 mmol g$^{-1}$ resin, 0.03 mmol scale), using DMF (99.5% pure) as a solvent. The coupling was carried out twice for each natural amino acid (4 equiv.) using HBTU (4 equiv.), HOBt (4 equiv.) and DIPEA (6 equiv.). The coupling of unnatural amino acids (2 equiv.) was performed once using HATU (2 equiv.) and DIPEA (4 equiv.). Fmoc groups were removed using a 20% v/v solution of piperidine in DMF. Washing steps were performed with DMF. Fmoc-NH-[PEG]2-CH$_2$COOH was purchased from Iris Biotech GmbH. Fmoc-NH-[PEG]24-CH$_2$CH$_2$COOH was purchased from PurePEG.

**Conjugation of fluorescein to N-terminus.** 5(6)-FAM (2 equiv.), DCC (2 equiv.) and HOBt (4 equiv.) in 3 ml DMF was shaken for 1 h.

**Peptide acylation with fatty acids.** Dde-lysine was selectively deprotected by incubating the resin three times with 2% hydrazine hydrate in DMF for 3 min

(400 r.p.m.), which also cleaved the Fmoc groups. Therefore, the N-terminus was protected by Boc in peptides not modified with fluorescein. Peptides were acylated at the lysine side chain by the addition of free fatty acid (2 equiv.), DCC (2 equiv.) and HOBt (4 equiv.) in 3 ml DMF and shaking at 400 r.p.m. for 1 h. Myristic acid, palmitic acid and stearic acid were purchased from Tokyo Chemical Industry (TCI).

**Peptide cleavage and deprotection.** Final peptides were deprotected and cleaved from the resin under reducing conditions (90% TFA, 2.5% H$_2$O, 2.5% thioanisole, 2.5% phenol, 2.5% EDT) for 4 h with shaking at RT. The resin was removed by vacuum filtration, and the peptides were precipitated with cold diethyl ether (50 ml) and incubation for 30 min at − 20 °C, and pelleted by centrifugation. The precipitated peptides were washed twice with cold diethyl ether (30 and 20 ml, respectively). Finally, peptides were either lyophilized or directly purified.

**Chemical cyclization.** Crude peptide (9 ml, 1.2 mM) was dissolved in 40% v/v MeCN and 60% v/v aqueous buffer (60 mM NH$_4$HCO$_3$, pH 8.0) and cyclization linker 1,3,5-tris-(bromomethyl)benzene (TBMB) or 1,3,5-triacryloyl-1,3,5-triazi-nane (TATA) was added (1 ml, 15 mM in MeCN). The final reagent and solvent concentrations in the reaction mixture were 1 mM peptide, 1.5 mM cyclization linker, 50% v/v aqueous buffer, and 50% v/v MeCN. The reaction mixture was incubated at 30 °C in a water bath for 1 h and then lyophilized.

**Peptide purification.** Peptides were purified by semi-preparative reversed-phase HPLC (PrepLC 4000-Waters system, Vydac C18 TP1022, 250 × 22 mm, 10 μm) using a flow rate of 20 ml min$^{-1}$ and a linear gradient of 30-90% v/v solvent B in 30 min (A: 99.9% v/v H$_2$O and 0.1% v/v TFA; B: 99.9% v/v MeCN and 0.1% v/v TFA). Fractions containing the desired peptide were lyophilized. Short peptides acylated with fatty acid were purified with a linear gradient of 50–90% v/v solvent B in 15 min.

**HPLC and mass spectrometric analysis.** The mass of purified peptides was determined by electrospray ionization mass spectrometry (ESI-MS) in positive ion

mode on a single quadruple liquid chromatograph mass spectrometer (LCMS-2020, Shimadzu). The purity was assessed by RP-HPLC (Agilent 1260 HPLC system) using a C18 column (Agilent ZORBAX 300SB-C18, 4.6 mm × 250 mm, 5 μm). Peptides were run at a flow rate of 1 ml min$^{-1}$ with a linear gradient of solvent B over 15 min (A: 94.9% v/v H$_2$O, 5% v/v MeCN and 0.1% v/v TFA; B: 94.9% v/v MeCN, 5% v/v H$_2$O and 0.1% v/v TFA).

**Fluorescence polarization albumin binding assay.** Human albumin (A1653), rat albumin (A6272), rabbit albumin (A0639) and human serum (H4522) were purchased from Sigma-Aldrich. Human albumin, rat albumin, rabbit albumin and human serum were serially diluted (twofold) in PBS (10 mM Na$_2$HPO$_4$, 1.8 mM KH$_2$PO$_4$, pH 7.4, 137 mM NaCl, 2.7 mM KCl, 0.01% v/v Tween-20). The highest concentrations were 28 μM for human albumin, 26 μM for rat albumin, 29 μM for rabbit albumin and 25 μM for human albumin in diluted serum (assuming an albumin concentration of 0.6 mM in serum). Forty-five microlitre of the dilutions were added to 15 μl of fluorescent peptide (100 nM) in black 96-well half area microtitre plates. The concentrations of albumin in the assay plate were ranging from 21 μM (human albumin), 19.5 μM (rat albumin), 21.75 μM (rabbit albumin) and 18.75 μM (human albumin from serum) to ~40 nM. The final concentration of fluorescent peptide was 25 nM in all wells. Fluorescence anisotropy was measured with an Infinite M200Pro plate reader (Tecan) using an excitation filter at 485 nm and an emission filter at 535 nm. Dissociation constant ($K_D$) were determined by nonlinear regression analysis of anisotropy versus the total concentration of proteins [P]$_T$ using the following equation:

$$A = A_f + (A_b - A_f)\left\{ \frac{\left[ ([L]_T + K_D + [P]_T) - \sqrt{([L]_T + K_D + [P]_T)^2 - 4[L]_T[P]_T} \right]}{2[P]_T} \right\} \quad (1)$$

$A$ is the experimental fluorescence anisotropy. $A_f$ and $A_b$ are the fluorescence anisotropy for the free and the fully bound fluorescent ligand, respectively. $[L]_T$ and $[P]_T$ represent the total fluorescent ligand and protein concentration, respectively. $K_D$ is the dissociation constant for the binding equilibrium. Values were calculated with Prism 5 (GraphPad software).

**Solubility assay.** The molar extinction coefficients of tagged peptides were determined at 205, 210 and 220 nm by measuring the absorption on a ultraviolet–visible spectrophotometer (GE Healthcare) in water. The solubility was measured by dissolving 10 mg of peptide in 1 ml PBS (10 mM Na$_2$HPO$_4$, 1.8 mM KH$_2$PO$_4$, pH 7.4, 137 mM NaCl, 2.7 mM KCl). A maximal amount of peptide was dissolved by inverting the tube at room temperature for 45 min. Non-dissolved peptide was removed by centrifugation for 30 min at 13,000 g and transfer of the supernatant to a new tube. This process was repeated once. The maximum solubility was determined by measuring the absorption at 205, 210 and 220 nm. The solubility of fluorescein labelled conjugates was determined by measuring the absorption at 495 nm and using a molar extinction coefficient of 68,000 M$^{-1}$ cm$^{-1}$.

**Protease inhibition assays.** $K_i$ values of UK18 and tagged UK18 were determined by measuring residual activity of recombinant human urokinase[19] (1.5 nM) in the presence of different dilutions of inhibitor (twofold dilutions, ranging from 1 μM to 0.97 nM final concentration) using the fluorogenic substrate Z-Gly-Gly-Arg-AMC (50 μM final concentration). $K_i$ values of PK128 and tagged PK128 were determined by measuring residual activity of human plasma kallikrein (0.25 nM) in presence of different dilutions of inhibitor (twofold dilutions, ranging from 250 to 0.24 nM final concentration) using the fluorogenic substrate Z-Phe-Arg-AMC (50 μM final concentration). The enzymatic assays were performed at 25 °C in a volume of 150 μl of assay buffer containing 10 mM Tris-Cl, pH 7.4, 150 mM NaCl, 10 mM MgCl$_2$, 1 mM CaCl$_2$, 0.01% v/v Tween-20, 0.1% (w/v) casein (bovine milk; C7078) and 5% v/v DMSO. Activity in presence of human albumin (99% pure, fatty acid free; A4327) was measured at an albumin concentration of 25 μM.

$K_i$ values of FXIIa inhibitor and tagged FXIIa inhibitor were determined by measuring residual activity of human β-FXIIa (0.4 nM) in the presence of different dilutions of inhibitor (twofold dilutions, ranging from 31.25 to 0.31 nM final concentration) using the fluorogenic substrate Boc-Gln-Gly-Arg-AMC (50 μM final concentration). The enzymatic assay was performed at 25 °C in a volume of 150 μl of assay buffer containing 10 mM Tris-Cl, pH 7.4, 150 mM NaCl, 10 mM MgCl$_2$, 1 mM CaCl$_2$, 0.01% v/v Tween-20, 0.1% (w/v) casein and 1% v/v DMSO. Activity in presence of human albumin (99% pure, fatty acid free; A4327) was measured at an albumin concentration of 10 μM and an increased concentration of FXIIa (3 nM) as fluorogenic substrate hydrolysis by albumin preparations interfered with the assay.

Fluorescence intensity was measured for 30 min or 1 h using an Infinite M200Pro plate reader (Tecan; AMC-substrates: excitation at 368 nm, emission at 467 nm) with a read every minute. Sigmoidal curves were fitted to the data using the following four-parameter logistic equation used for dose–response curves

$$y = \frac{100}{1 + 10^{(\log IC_{50} - x)p}}, \quad (2)$$

wherein $x$ is the peptide concentration, $y$ is the residual percentage of protease

activity, $p$ is the Hill slope. IC$_{50}$ values were derived from the fitted curve using Prism 5 (GraphPad software). The $K_i$ values were calculated based on the IC$_{50}$ values using the Cheng–Prusoff equation

$$K_i = \frac{IC_{50}}{1 + \frac{[S]_0}{K_m}}, \quad (3)$$

wherein IC$_{50}$ is the functional strength of the inhibitor, $[S]_0$ is the total substrate concentration, and $K_m$ is the Michaelis–Menten constant. The following $K_m$ values were used for the calculation: 112 μM for urokinase using Z-Gly-Gly-Arg-AMC[19], 165 μM for plasma kallikrein using Z-Phe-Arg-AMC[22], 256 μM for FXIIa using Boc-Gln-Gly-Arg-AMC[25].

The specificity of tagged FXIIa inhibitor was assessed by measuring the inhibition of the following human serine proteases: urokinase 1.5 nM, factor XIa 0.25 nM, plasma kallikrein 0.25 nM, thrombin 1 nM, plasmin 2.5 nM and trypsin 0.1 nM. The following fluorogenic substrates were used at 50 μM final concentration: Z-Phe-Arg-AMC for plasma kallikrein, Boc-Phe-Ser-Arg-AMC for factor XIa, Z-Gly-Gly-Arg-AMC for urokinase, thrombin and trypsin, and H-D-Val-Leu-Lys-AMC for plasmin. Enzymes were purchased from Molecular Innovations or Innovative Research.

**Plasma stability assay.** An amount of 20 μM of peptide was incubated in human plasma (Innovative Research) at 37 °C in a water bath. Probes were taken at different time points (5 min, 0.5, 1, 2, 4, 8, 24 and 48 h), diluted to 3 μM (urokinase inhibitor) or 0.75 μM (plasma kallikrein inhibitor) with assay buffer (10 mM Tris-Cl, pH 7.4, 150 mM NaCl, 10 mM MgCl$_2$, 1 mM CaCl$_2$, 0.01% v/v Tween-20, and 0.1% (w/v) casein), heat inactivated by incubation at 65 °C for 20 min, centrifuged at 16,000 g for 5 min and the supernatant transferred to a new tube. The inhibitory activity of each sample was assessed by preparing a series of twofold dilutions and measuring residual protease activity for urokinase and plasma kallikrein as described above. Residual inhibition in % was calculated using the equation IC$_{50,0h}$/IC$_{50,xh}$*100, wherein IC$_{50}$,0 h is the functional strength of the inhibitor at time point 0 and IC$_{50,xh}$ the functional strength of inhibitor after one of the different plasma incubation periods mentioned above. The resulting percentages of inhibition were plotted in Prism 5 software (GraphPad software) versus time and analysed using a one-phase decay model.

**Pharmacokinetic study in rats.** Female Sprague Dawley rats were injected with 0.5 mg kg$^{-1}$ tag-UK18, 0.5 mg kg$^{-1}$ tag-PK128, 0.3 mg kg$^{-1}$ UK18 or 0.25 mg kg$^{-1}$ tag in 200 μl PBS, pH 7.4, 0.01% SDS (corresponding to around 130 μM) via the tail vein (three animals per group). Blood samples (100 μl) were collected at different time points via the orbital sinus into tubes containing EDTA, centrifuged 360 g for 15 min at 4 °C and the plasma stored at −80 °C. Peptide injection and blood sample collection was performed by Washington Biotech Inc. following ethical standards for animal studies of the Office for Laboratory Animal Welfare (OLAW), a division of the US Public Health Service as administered by the National Institutes for Health. The samples were analysed by RP-HPLC (Agilent 1260 HPLC system) equipped with a fluorescence detector (Shimadzu RF-10AXL detector, excitation at 445 nm, emission at 535 nm) using a C18 column (Agilent ZORBAX 300SB-C18, 4.6 mm × 250 mm, 5 μm) and a linear gradient of solvent B 0–100% v/v over 15 min at a flow rate of 1 ml min$^{-1}$ (solvent A: 94.9% v/v H$_2$O, 5% v/v MeCN and 0.1% v/v TFA; solvent B: 94.9% v/v MeCN, 5% v/v H$_2$O and 0.1% v/v TFA). Plasma samples were processed before RP-HPLC analysis as follows. 2 μM fluorescein was added as standard to 5 μl plasma sample. 5 μl of 7 M guanidinium hydrochloride solution, pH 2 (adjusted with HCl) was added, mixed and incubated for 10 min at RT to denature proteins. Two hundred microlitre of cold ethanol was added and incubated on ice for 1 h to precipitate proteins. Samples were centrifuged at 9,000 g for 15 min at 4 °C and the supernatant dried by centrifugal evaporation under vacuum. Dried samples were dissolved by stepwise addition of 10 μl DMSO and 90 μl solvent A (99.9% v/v H$_2$O, 0.1% v/v TFA). 95 μl were analysed by RP-HPLC. Peptides were quantified by integrating the area under the peaks, and normalization was based on the standard and comparison to standard curves. Standard curves were obtained by analysing known peptide quantities that were dissolved in plasma and processed as well as analysed using the same procedure. Pharmacokinetic parameters were calculated using a two-compartment model.

**Coagulation assays.** Activated partial thromboplastin time was measured on a STart4 coagulation analyser (Diagnostica Stago). One hundred microlitre of rabbit plasma was incubated at 37 °C with 100 μl of Dade/Actin (contains phospholipids from rabbit brain and ellagic acid, Siemens) for 3 min. One hundred microlitre of human plasma was incubated at 37 °C with 100 μl of Pathromtin* SL (silicon dioxide particles, plant phospholipids in 4-(2-hydroxyethyl)-1-piperazineethanesulfonic acid (HEPES) buffer system, Siemens) for 2 min. Coagulation was triggered by addition of 100 μl of CaCl$_2$ solution (25 mM; Siemens). The time to coagulation was assessed by monitoring the movement of a steel ball in the plasma.

**Pharmacokinetic study in rabbits.** Female New Zealand White rabbits were injected with 3.7 mg kg$^{-1}$ FXIIa inhibitor in 1 ml PBS, pH 7.4 (3 mM) or

5 mg kg$^{-1}$ tag-3xPEG24-FXIIa inhibitor in 3 ml PBS, pH 7.4, 0.01% SDS (570 µM) via the ear vein (three animals per group). Blood samples (1.8 ml) were collected at different time points via the auricular artery of the opposite ear into tubes containing Na citrate, centrifuged 360 g for 15 min at 4 °C and the plasma stored at −80 °C. Peptide injection and blood sample collection were performed by Washington Biotech Inc. following ethical standards for animal studies of the Office for Laboratory Animal Welfare (OLAW), a division of the US Public Health Service as administered by the National Institutes for Health. Plasma samples were treated as the rat plasma samples described above. Samples from rabbits treated with tag-3xPEG24-FXIIa were analysed by RP-HPLC using fluorescence detection as described above. Samples from rabbits treated with FXIIa inhibitor were analysed on a single quadrupole liquid chromatography mass spectrometer (LC/MS-2020, Shimadzu), using a C18 column (Phenomenex Kinetex 2.6 µm C18 100 Å, LC Column 50 × 2.1 mm), a linear gradient of solvent B (5–30%) over 4.5 min at a flow rate of 1 ml min$^{-1}$ (solvent A: 94.95% v/v H$_2$O, 5% v/v MeCN and 0.05% v/v HCOOH; solvent B: 94.95% v/v MeCN, 5% v/v H$_2$O and 0.05% v/v HCOOH) and ESI detection in positive mode. Peptide analysed by RP-HPLC and fluorescence detection was quantified as in the rat experiment described above. Peptide analysed by liquid chromatography-mass spectrometry was quantified based on the absolute intensities of the detected mass peaks (M$^{3+}$ and M$^{4+}$) and adjusted according to the intensity of the mass peaks of the internal standard. Peptide concentrations in the plasma samples were calculated according to linear external standard curves obtained from dilution series of the peptide prepared in rabbit plasma. Pharmacokinetic parameters were calculated using a two-compartment model.

**Data availability.** Data supporting the findings of this study are available within the article and its Supplementary Information File and from the corresponding author upon reasonable request.

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

## Acknowledgements

This work was supported by the Swiss National Science Foundation, grant no. 157842.

## Author contributions

A.Z. and C.H. conceived the strategy, designed the experiments, analysed the data and wrote the manuscript. A.Z. performed all experiments. S.J.M. and J.W. established the blood coagulation assays and the rabbit *in vivo* study, and contributed to the writing of the manuscript. K.D. contributed to the writing of the manuscript.

## Additional information

**Competing interests:** C.H. is a scientific founder of Bicycle Therapeutics. The remaining authors declare no conflict of interests.

