## [Peer Review File · Nature Communications]

REVIEWERS' COMMENTS:

Reviewer #1 (Remarks to the Author):

The authors have pursued enhancement of circulatory lifetime for small peptides by conjugating them to fatty acids such as myristic, palmitic, or stearic, thereby serving as weak-affinity ligands for serum albumin. Subsequent studies focused on palmitic acid, as it had the highest albumin binding affinity. An iterative optimization process was followed to identify peptide attachment sequence contexts for the palmitic acid attachment, and a sequence with 37x improved albumin binding affinity was identified. Attachment to peptidic payloads lowered both their efficacy and the affinity for albumin, but nevertheless led to significant increases in circulating drug exposure. This work is a useful demonstration of the importance of the often-ignored bridging structures between bispecific constructs such as these.

Reviewer #2 (Remarks to the Author):

This paper describes a novel approach to enhancing drug action by preparing conjugates which bind to the natural plasma carrier protein, albumin, via a combination of hydrophobic and H-bond interactions. Thus, although the concept of using albumin to carry synthetic drug molecules is long-established, in this case the novelty arises from the design of the albumin-binding domain in the conjugate pro-drug.

This is a very well-written manuscript and very easy to read. As a reviewer it is very rewarding to see that the authors have taken great care with controls, and considered biological reproducibility in their experiments (e.g. Fig S18 when the data for 3 independent animals are shown as 3 independent experiments, with actually a surprisingly low inter-sample variation). This quality, when combined with the interesting and novel idea, makes this manuscript suitable for Nature Communications, with very few revisions necessary.

Clarification

Albumin is known to have seven fatty acid binding sites, which vary in affinity based on whether other fatty acids are bound, and of these there are reported to be 3 high and 4 lower affinity binding sites. The authors' strategy in this paper, i.e. to generate strong albumin binders which are not highly amphiphilic, was to flank the lysine bearing the palmitoyl derivative with serines or glycines, and then iteratively replace individual amino acids with analogues. This generated peptides ranging

from SSS-K-(palmitoate)-SSS to EYE-K-(palmitoate)-EYE, with the latter showing the highest binding affinity. It would improve the manuscript if the authors were to include a short discussion as to the likely binding interactions of these amino acids around the fatty acid pockets, and to indicate which region of the albumin, i.e. which of the 7 fatty acid binding sites and associated albumin amino acid residues, are likely associated with EYE-K-(palmitoate)-EYE. Do the most effective binders match complementary sequences in the albumin PDB structures? If not, what are the origins of the flanking peptide-protein interactions?

Minor revisions

Abbreviations used should be defined – e.g for the non organic chemists it is not easy to work out what are TBMB or TATA.

What curve fitting software was used for the inhibitory assays? In Figures S8 and S15, some of the fits look very approximate, and errors in activity constants could easily have been introduced in, for example, S15a for the Factor XIIa inhibition.

Some recent papers related to this area should be cited: ACS Macro Lett., 2016, 5 (10), 1089–1094, Chem. Commun. 2014, 50, 6394– 6397

Reviewer #3 (Remarks to the Author):

This paper describes the discovery and optimization of a lipid-modified bicyclic peptide with very high affinity for serum albumin, and its use as a “tag” to stabilize cargo toward metabolic degradation in vivo. Covalent attachment of the tag to a previously discovered FXIIa inhibitor improved the half-life of the inhibitor 24-fold over the untagged inhibitor, and pharmacological efficacy (in an ex-vivo coagulation model) was observed for the conjugate but not the untagged molecule.

This paper presents an interesting and potentially useful technical advance in the use of serum albumin to increase the serum half-life of peptidic drugs, but it does not represent a fundamental breakthrough. Lipidation and the use of stabilizing peptides are common methods for achieving albumin binding, and the list of drugs that are either approved or in clinical development which utilize non-covalent binding to serum albumin to increase plasma stability is relatively long (see Biochim Biophys Acta 1830 (2013) 5526–5534). Many of these examples show even greater fold-enhancement in stability (with tag vs. without) compared to the examples in the current manuscript.

While the bicyclic peptides presented in this paper are more soluble and easier to synthesize than some of the current tags, these features alone do not (in my opinion) reach the threshold of “extreme importance to scientists in the specific field” as stated in the “criteria for publication” found in the Nature journals’ guidelines for reviewers. These tag-FXIIa inhibitor conjugates provide a promising advance in anti-thrombotic therapy, and the tags themselves provide an additional tool in the array of methods for increasing plasma half-life. Therefore, I recommend publication in a medicinal chemistry journal such as *Eur. J. Med. Chem.*, *J. Med. Chem.*, or *ACS Med. Chem. Lett.*

Point-by-point discussion of reviewers' comments:

Reviewer #1 (Remarks to the Author):

The authors have pursued enhancement of circulatory lifetime for small peptides by conjugating them to fatty acids such as myristic, palmitic, or stearic, thereby serving as weak-affinity ligands for serum albumin. Subsequent studies focused on palmitic acid, as it had the highest albumin binding affinity. An iterative optimization process was followed to identify peptide attachment sequence contexts for the palmitic acid attachment, and a sequence with 37x improved albumin binding affinity was identified. Attachment to peptidic payloads lowered both their efficacy and the affinity for albumin, but nevertheless led to significant increases in circulating drug exposure. This work is a useful demonstration of the importance of the often-ignored bridging structures between bispecific constructs such as these.

Reviewer #2 (Remarks to the Author):

This paper describes a novel approach to enhancing drug action by preparing conjugates which bind to the natural plasma carrier protein, albumin, via a combination of hydrophobic and H-bond interactions. Thus, although the concept of using albumin to carry synthetic drug molecules is long-established, in this case the novelty arises from the design of the albumin-binding domain in the conjugate pro-drug.

This is a very well-written manuscript and very easy to read. As a reviewer it is very rewarding to see that the authors have taken great care with controls, and considered biological reproducibility in their experiments (e.g. Fig S18 when the data for 3 independent animals are shown as 3 independent experiments, with actually a surprisingly low inter-sample variation). This quality, when combined with the interesting and novel idea, makes this manuscript suitable for Nature Communications, with very few revisions necessary.

Clarification

Albumin is known to have seven fatty acid binding sites, which vary in affinity based on whether other fatty acids are bound, and of these there are reported to be 3 high and 4 lower affinity binding sites. The authors' strategy in this paper, i.e. to generate strong albumin binders which are not highly amphiphilic, was to flank the lysine bearing the palmitoyl derivative with serines or glycines, and then iteratively replace individual amino acids with analogues. This generated peptides ranging from SSS-K-(palmitoate)-SSS to EYE-K-(palmitoate)-EYE, with the latter showing the highest binding affinity. It would improve the manuscript if the authors were to include a short discussion as to the likely binding interactions of these amino acids around the fatty acid pockets, and to indicate which region of the albumin, i.e. which of the 7 fatty acid binding sites and associated albumin amino acid residues, are likely associated with EYE-K-(palmitoate)-EYE. Do the most effective binders match complementary sequences in the albumin PDB structures? If not, what are the origins of the flanking peptide-protein interactions?

Re: This is an interesting question. We have now looked at structures of albumin complexed with albumin-binding proteins to see which amino acids form key interactions with albumin, and if there is a similarity to the peptide sequence that we had evolved. Unfortunately, we have not found any obvious correlation. We have now added a sentence to report this finding:

"The evolved amino acid sequence did not share any homology with binding regions of natural albumin-binding proteins."

Minor revisions

Abbreviations used should be defined – e.g for the non organic chemists it is not easy to work out what are TBMB or TATA.

Re: We have now ensured that all uncommon abbreviations are properly introduced when they are used the first time, including TBMB and TATA.

What curve fitting software was used for the inhibitory assays? In Figures S8 and S15, some of the fits look very approximate, and errors in activity constants could easily have been introduced in, for example, S15a for the Factor XIIIa inhibition.

Re: We have used the software Prism from GraphPad to fit sigmoidal dose response curves to the data. The software and a four-parameter logistic equation used is now described in detail in the Materials and Methods section.

The fitting of two data sets is indeed not ideal in one panel of Supplementary Fig. 8 (panel b, tag-UK18, with albumin, green curves) and two panels of Supplementary Fig. 15 (FXIIa inhibitor with albumin, red curves). We have improved the fitting by not forcing the curves to start at 100% on the y-axis. This gave perfect fits. We found additionally the same problem for Supplementary Fig. 18 and fixed this as well. The newly calculated K_i values did not differ much from those obtained with the previous fitting. We have nevertheless transferred the new values to Table 1 and Supplementary Table 1.

Some recent papers related to this area should be cited: ACS Macro Lett., 2016, 5 (10), 1089–1094, Chem. Commun. 2014, 50, 6394– 6397

Re: The two studies mentioned are interesting and nice examples of recently developed albumin-based - drug delivery strategies that rely on covalent coupling of drugs to albumin. Instead of citing individual examples in this field, we chose to cite a recent review article that covers studies relying on covalent conjugation of drugs to albumin: Bern, M. *et al.*, The role of albumin receptors in regulation of albumin homeostasis: implications for drug delivery. J. Controlled Release (2015).

Reviewer #3 (Remarks to the Author):

This paper describes the discovery and optimization of a lipid-modified bicyclic peptide with very high affinity for serum albumin, and its use as a “tag” to stabilize cargo toward metabolic degradation in vivo. Covalent attachment of the tag to a previously discovered FXIIa inhibitor improved the half-life of the inhibitor 24-fold over the untagged inhibitor, and pharmacological efficacy (in an ex-vivo coagulation model) was observed for the conjugate but not the untagged molecule.

This paper presents an interesting and potentially useful technical advance in the use of serum albumin to increase the serum half-life of peptidic drugs, but it does not represent a fundamental breakthrough. Lipidation and the use of stabilizing peptides are common methods for achieving albumin binding, and the list of drugs that are either approved or in clinical development which utilize non-covalent binding to serum albumin to increase plasma stability is relatively long (see Biochim Biophys Acta 1830 (2013) 5526–5534). Many of these examples show even greater fold-enhancement in stability (with tag vs. without) compared to the examples in the current manuscript.

Re: None of the reported albumin-binding ligands has all the desired properties (affinity, solubility, ease of synthesis). In contrast, the ligand that we have newly developed combines all these properties.

While the bicyclic peptides presented in this paper are more soluble and easier to synthesize than some of the current tags, these features alone do not (in my opinion) reach the threshold of “extreme importance to scientists in the specific field” as stated in the “criteria for publication” found in the Nature journals’ guidelines for reviewers. These tag-FXIIa inhibitor conjugates provide a promising advance in anti-thrombotic therapy, and the tags themselves provide an additional tool in the array of methods for increasing plasma half-life. Therefore, I recommend publication in a medicinal chemistry journal such as Eur. J. Med. Chem., J. Med. Chem., or ACS Med. Chem. Lett.